# Built-In Packaging for Single Terminal Devices

**DOI:** 10.3390/s22145264

**Published:** 2022-07-14

**Authors:** Ahmet Gulsaran, Bersu Bastug Azer, Samed Kocer, Sasan Rahmanian, Resul Saritas, Eihab M. Abdel-Rahman, Mustafa Yavuz

**Affiliations:** 1Mechanical and Mechatronics Engineering Department, University of Waterloo, Waterloo, ON N2L 3G1, Canada; bbastuga@uwaterloo.ca; 2Waterloo Institute for Nanotechnology (WIN), University of Waterloo, Waterloo, ON N2L 3G1, Canada; skocer@uwaterloo.ca (S.K.); resulsaritas@gmail.com (R.S.); eihab@uwaterloo.ca (E.M.A.-R.); 3Systems Design Engineering Department, University of Waterloo, Waterloo, ON N2L 3G1, Canada; s223rahm@uwaterloo.ca

**Keywords:** packaging, wire bonding, actuator application, micro resonator, sensor application, MEMS, NEMS

## Abstract

An alternative packaging method, termed built-in packaging, is proposed for single terminal devices, and demonstrated with an actuator application. Built-in packaging removes the requirements of wire bonding, chip carrier, PCB, probe station, interconnection elements, and even wires to drive single terminal devices. Reducing these needs simplifies operation and eliminates possible noise sources. A micro resonator device is fabricated and built-in packaged for demonstration with electrostatic actuation and optical measurement. Identical actuation performances are achieved with the most conventional packaging method, wire bonding. The proposed method offers a compact and cheap packaging for industrial and academic applications.

## 1. Introduction

The typical working principle of a sensor is two-fold. First, the system is set into the sensing state that is sensitive to the target input. Generally, an external signal is required for this activation, except for self-actuated structures [1]. Second, the output signal containing detection information is translated into the proper form. The majority of the sensor research focuses between these two feedthroughs where detection interaction happens by exploring state-of-the-art sensing concepts [2], functionalizing high-performance materials [3], and expanding fabrication limits [4]. Developments in these areas enabled a better understanding of the universe by measuring quantized unit of electrical conductance [5] and mass of Higgs boson [6] or resulted in detecting more practical targets such as the mass of a single proton [7] or energy of a single photon [8] under ideal conditions. As sensitivity increases, the signal levels decrease, and feedthrough losses of these tiny messages start to play a more significant role. Therefore, developments in the sensing state need to be followed by the ones in feedthrough for keeping the sensing state precisely and reading the output signal with minimal losses.

Principally, a package is required for interfacing the chip with actuation and readout circuits, and there are three popular methods for bonding a chip on top of a carrier board. In wire bonding, the chip is attached to the package with welded fine wire leads, as shown in Figure 1b. In flip-chip bonding, the top layer of the chip is soldered with the carrier, as illustrated in Figure 1c. In tape-automated bonding, a thin conductor tape is attached on top of the metal pads of the device, as demonstrated in Figure 1d. Each of these techniques introduces design restrictions for compatibility. Although all the components and processes are well standardized, they introduce additional feedthrough loss, complexity, size, and cost. In addition, it is known that feedthrough noises and losses play a significant role in the electrical actuation and measurements of MEMS and NEMS [9,10,11,12]. In addition modeling of the parasitic effects of these noise sources was attempted, to minimize the damage [13].

Instead of putting up with these problems, it is aimed to eliminate the source directly by proposing a new concept called built-in packaging. Unlike all conventional methods, in the proposed method, the built-in packaging, a connector is directly attached to the die without any package or carrier board, as illustrated in Figure 1e. The main goal of this work is to eliminate all the interconnection components, even wires, and facilitate the device into a plug-and-play operation.

The first-ever demonstration of the proposed concept is performed with a single-terminal device for the sake of simplicity. A single-terminal resonator is designed, modeled, fabricated, packaged, and characterized. The one and only electrical terminal is used on the actuation side, and the mechanical motion is detected optically. Therefore, the actuation performance of the proposed method is compared with the most conventional method, the wire bonding, to inspect the packaging performance. The actuation and readout schemes are illustrated in Figure 2. To examine the performance of the operation, the results are compared with the most conventional method, wire bonding.

Though only a single-terminal actuator application is illustrated in this work, the proposed concept is valid for sensor applications (accelerometer [14], gyroscope [15], mass sensor [16], humidity sensor [17], temperature sensor [18], pressure sensor [19], gas sensor [20], water sensor [21], magnetometers [22], photoacoustic sensors [23], FET-biosensor [24], and permittivity sensor [25]), optical and photonic applications (micromirror [26], microswitch [27], LiDAR [28], beam steering [29], bolometer [30], photon detector [31], and PeCOD [32]), RF applications (diode [33], transistor [34], antennas [35], switch [36], phase shifter [37], filter [38], and tunable capacitor [39]), microfluidics (micropump [40], microdroplet generator [41], and microvalve [42]), and other applications (energy harvester [43], rectenna [44], gripper [45], etc.).

## 2. Actuator Design

To illustrate the proposed packaging method for an actuator application, a microresonator needed to be designed, and this design should be compatible with the actuation and characterization setups, and easy to fabricate and package. First, the important criteria were specified and then associated with geometrical and material properties, and finally, a complete design satisfying these criteria was developed. In addition, a mathematical model was built and numerically solved for optimization and verification. Detailed equipment information is provided in Appendix A.

### 2.1. Compatibility

While it is possible to make in-plane motion measurements with LDV [46], out-of-plane motion is easier to catch. In addition, the lowest eigenmode offers the highest deformation and results in a larger output signal from the LDV. Thus, the main operation mode of the resonator was chosen as the first out-of-plane mode. Both cantilever and fixed–fixed structures are appropriate for this purpose and fixed–fixed boundary conditions were preferred to minimize the curvature.

LDV converts motion into an electrical signal from the Doppler shift in the reflected laser beam frequency due to the movement of the surface. The laser spot has a finite diameter, called airy disk, and should fully cover the top surface of the resonator, otherwise the laser beam might also reflect from stationary parts, causing the output signal to decay. This can especially affect the calibration between motion and electrical signals and might cause imprecise results. To eliminate all these complications, the minimum width of the resonator, b_min_, should be larger than the spot size of the beam. The spot size of the beam is defined as the Airy disk diameter, $d_{\mathrm{airy}}$, in Equation (1) [47].
(1)$$d_{\mathrm{airy}}=1.22\frac{\lambda }{\mathrm{NA}}$$

λ and NA are the wavelength of the laser beam and numerical aperture of the objective lens, respectively. Smaller spots can be achieved by using an objective lens with higher NA or a laser beam with a lower wavelength. In our setup, λ=633 nm and NA=0.42 yield a theoretical limit of $d_{\mathrm{theoretical}}=1.8\;\mu m$. On the other hand, imperfect calibration introduces optical aberration, and the practical spot diameter is around d≅4 μm; hence, the width of the resonator should be wider than 4 μm.

Electrical conductance is required to both transmit voltage and generate electrostatic force. Moreover, it is desired to have high reflectance on the resonator surface to obtain a high output from the LDV. The relationship between optical reflectance, R, and electrical conductivity, σ, for conductive materials is well-defined with Hagen–Rubens formula in Equation (2) [48].
(2)$$R\approx 1-4\sqrt{\frac{\pi \epsilon_{0}}{\sigma \lambda }}$$

$\epsilon_{0}$ and λ are the vacuum permittivity and optical wavelength, respectively. This relation shows that materials with high electrical conductivity also have high optical reflectivity. Therefore, a conductor material is preferred for the resonating medium. In terms of fabrication, it should be easy to etch for patterning, compatible with Al wire bonding, and cheap to deposit. Hence, Al was selected as the structure layer after considering all these concerns.

Larger actuation force produces a larger output signal and increases the signal-to-noise ratio. By neglecting the fringing field, the electrostatic force per unit length applying on the undeflected structure, $F_{\mathrm{es}}$, is given in Equation (3) [49].
(3)$$F_{\mathrm{es}}=\frac{\epsilon {\;b\;V}^{2}}{{2d}^{2}}$$

ϵ, b, V, and d are the dielectric constant of the gap medium, width of the resonator, actuation voltage, and gap width, respectively. The actuation voltage is limited by the function generator. Although decreasing the air gap between the resonator and actuation electrode increases the electrostatic force, it could introduce additional squeezed film damping, $c_{\mathrm{sf}}$, in Equation (4) [50].
(4)$$c_{\mathrm{sf}}=f\left(\frac{b}{l}\right)\frac{{\mu \;l}^{4}}{d^{3}}$$

$f\left(\frac{b}{l}\right)$, l, and μ are a geometrical factor, the length of the resonator and the viscosity of the gap medium, respectively. Moreover, decreasing the air gap width can also produce an underetch issue, discussed in Section 2.2. As a result, the stiffness of the beam needs to be as weak as possible to convert limited electrostatic force into the greatest deflection. From a geometrical point of view, this can be achieved by having a large length-to-thickness ratio (>100), and the exact ratio was calculated as 160 using the simulations.

The dimensions in MEMS and NEMS tend to scale down, and natural frequencies increase according to the scaling law [51]. Therefore, it is desired to push the limits of frequency while designing the actuator to demonstrate the proposed method for the most updated and future applications. Furthermore, a higher frequency regime offers better quality factors. Nevertheless, the digital velocity output of our LDV starts decaying after 1.5 MHz, and the natural frequency should exceed this threshold for equivalent measurement. Since the geometrical properties, material properties, and residual stress affect the natural frequency, mathematical models were simulated, and the length and thickness were determined as 40 μm and 250 nm, respectively.

### 2.2. Producibility

Standard optical lithography techniques allow patterning of the resonator with the desired length, and width [52]. The releasing step is critical when etching the sacrificial layer underneath the resonator because of aggressive etchants and sticking problems due to surface tension forces. To eliminate these problems, a polymer-based sacrificial layer was chosen with dry release in oxygen plasma [53]. However, plasma might struggle to reach underneath the resonator, especially if the width-to-air-gap ratio is high and can cause saturated underetch. Therefore, a low width-to-air-gap ratio is desired for a smooth release process. The width was chosen as 4 μm, $b_{\min}$, and an air gap of 4 μm was chosen to obtain a unity ratio.

### 2.3. Simulation

The three main aims of running simulations are (i) to identify the design parameters by calculating the natural frequency of the beam under the presence of residual stress, (ii) to perform a failure analysis to define the working regime, and (iii) to obtain the mode shape to compare with the experimental results. An FEM (Finite Element Model) was constructed using COMSOL^®^ by following the AC/DC and Structural Mechanics Modules User’s Guide [54,55]. In addition to that, a ROM (Reduced Order Model) was built [49], and the details are provided in the Appendix A. In the numerical calculations, Young’s modulus, density, Poisson ratio, and tensile strength of Al are taken as 69 GPa, 2.71 g/cm^3^, 0.33, and 90 MPa, respectively [56].

The residual stress gets affected by so many parameters such as growth temperature and rate [57], material, deposition method, thickness [58], epitaxial mismatch [59], surface roughness, and contact angle [60]. Therefore, it is not feasible to make an accurate prediction by a simple calculation. Instead, a reasonable range, ±20 MPa, was defined based on the literature data [61,62,63], and the simulations were run in this region. The length and thickness were determined by keeping the first natural frequency below 1.5 MHz. The relationship between the residual stress and the first natural frequency of the finalized design is shown in Figure 3a. It should be noted that compressive stress larger than 10 MPa might cause buckling during the release step, which is not desired. Additionally, buckling stress determines the thermal budget to calculate temperature limits. Fortunately, the first natural frequency of the fabricated structure is obtained at 1.4 MHz, corresponding to tensile stress around 17 MPa.

Secondly, failure analyses were run to determine the maximum actuation voltage limit. While there have been several failure mechanisms that worked in the literature, only two main mechanisms were considered in this analysis [64]. As the actuation voltage increases, the beam deflects more, and after some critical deflection, the beam can stick to the stationary electrode or collapse due to tensile stress. The former phenomenon is called pull-in and can be either static [65] or dynamic [66]. The dynamic pull-in analysis requires specific initial conditions and extra effort to generalize. For the sake of simplicity, only the static case was considered in this analysis. Alternatively, the beam might collapse if von Mises stress exceeds ultimate tensile stress. The variation of deflection and von Mises stress with respect to the actuation voltage is shown in Figure 3b. When both failure mechanisms are considered, collapse happens earlier at a voltage level of around 200 V. To prevent failure, the maximum pulse voltage, 60 V, was set away from this value.

Finally, mode shape analyses were conducted to verify the experimental results and are illustrated in Section 6.3. 

## 3. Fabrication

The schematics of the fabrication process are shown in Figure 4 and Table 1 and 4″ prime grade p-doped Silicon wafer with a 50 nm-thick thermal oxide was chosen as the starting substrate.

A local actuator electrode was preferred for having a better electrical contact in packaging and eliminating excessive parasitic capacitance. It is patterned with a simple bilayer lift-off process, as shown in Figure 4a [67]. First, HMDS treatment was performed to promote the adhesion between the substrate and resist layer. The underlayer (PMGI SF7) and positive-tone UV-resist (Shipley S1805) were spin-coated and soft-baked, sequentially. The pattern was exposed to 405 nm wavelength light with MLA (maskless lithography aligner) and developed in MF-319. Then, 150 nm-thick Al film was deposited with e-beam (electron beam) deposition, and the excessive metal was lifted off.

The sacrificial layer, Figure 4b, was built by spin-coating a negative-tone photoresist (AZ nLOF 2035) to reach 4 μm thickness and hard-baking under vacuum to prevent outgassing.

To construct the structure layer illustrated in Figure 4c, firstly, 250 nm-thick Al film was deposited with e-beam deposition. Then, a positive-tone UV-resist (Shipley S1805) was spin-coated and soft-baked, and the design was exposed to 405 nm wavelength light with MLA and developed in MF-319. Excessive Al was dry etched with ICP-RIE (inductively coupled plasma—reactive ion etching).

The wafer was diced into smaller pieces with an automatic dicing. Finally, the structure was released by etching the sacrificial layer with a photoresist stripper, as shown in Figure 4d.

Optical microscope and SEM images of the fabricated resonator are shown in Figure 5. The laser spot almost covers the resonator as expected, as shown in Figure 5b.

## 4. Packaging

The images of the die with and without packaging are shown in Figure 6a. The middle electrode of the die is the actuation electrode, while the rest is the ground electrode. The experimental setup for the reference measurements with a probing technique from the die without packaging is shown in Figure 6b. To obtain electrical contact with the chip, a probe station, micromanipulators, and a relatively larger space are required.

### 4.1. Wire Bonding

The die was fixed to the chip carrier with double-sided tape and wire-bonded with 25 um Al wire using a semi-automatic wedge–wedge bonder (Westbond 4546E), as shown in the middle of Figure 6a. Then, the chip carrier was mounted on a custom PCB connecting the chip carrier to the testing equipment, and an image from the experimental setup is shown in Figure 6c.

### 4.2. Built-In

Unlike wire bonding, there is no equipment required for built-in packaging, and the die can be simply attached to the SMA (SubMiniature version A) connector jack, as shown in the right side of Figure 6a. A foam tape was placed under the die to tolerate the thickness difference between the die and connector and encourage assembly. An image from the experimental setup is shown in Figure 6d, and in contrast with wire bonding, the built-in package does not require any testing PCB and is ready for connection to the testing equipment. It also allows temporarily packaging, and the die can be easily dismounted from the SMA connector jack if required. In this work, the connector was removed after testing, and the die was stored in its carrier, separately. For long-term applications where permanent packaging is necessary, the assembly can be promoted with glue or soldering.

## 5. Characterization

### 5.1. Purpose

The main aim of characterization is to compare the proposed packaging idea, built-in, with the conventional one, wire bonding, in terms of actuating performance. It is desired to test the same die with both types of packaging to prevent any variation. Although built-in offers temporary packaging and is suitable for this idea, it is destructive and tedious to remove and package back wire bonding, due to destructing contact pads and wire-bonder tool requirements. Therefore, two similar dies were packaged separately with each method and characterized simultaneously. To overcome the slight variation between dies, both were also characterized under a probe station with micromanipulators as a reference, and the performance comparison was conducted with respect to this reference.

Initially, frequency sweep measurements were performed for AC analyses. Then, DC analyses were completed with pulse response measurements. In addition, to demonstrate the applications inside an isolated chamber, pulse response measurements under vacuum conditions were prepared. Finally, mode shapes were measured and compared with the theoretical ones to associate the experimental results with the theory.

The methodology mentioned in Figure 2 was followed for building the experimental setup for both measurements shown in Figure 7.

### 5.2. Frequency Sweep

The frequency sweep measurement setup is illustrated in Figure 7a. As there are two different chips, each one was measured with and without micromanipulators. The actuation signal was applied directly from the function generator in a linearly sweeping form at 20 V peak to peak and recorded through oscilloscope. The electrostatic force was related to the square of the actuation signal, so the mid frequency was chosen near half of the resonance frequency. For sensitive measurements over a wide range, three different sweeps were performed with 300 kHz, 150 kHz, and 25 kHz spans with 1 ms sweeping time. The laser beam of the LDV was focused on the center of the beam, and the analog velocity output signal of the LDV was measured from the oscilloscope in the time domain. To synchronize the phase of the input signal with the response, a trigger signal with the sweeping time was applied to perform a direct transformation from the time domain to the frequency domain.

The data read from the oscilloscope is shown in Figure 8a. Both the actuation and velocity signals were in the time domain and, firstly, transformed into the frequency domain. Then, the root mean square of the velocity with respect to the actuation frequency was obtained and plotted in the following section to obtain the Q-factor and resonance frequencies. Finally, the phase difference between the force and velocity was obtained in the frequency domain to phase behavior and obtain the Q-factor. The phase difference in the time domain is shown in Figure 8b. The calculations related to this part are illustrated in the Appendix A.

### 5.3. Pulse Response

In the pulse response tests, a 4 V peak-to-peak pulse signal with 15x amplification was applied from the function generator at 0.1% pulse width. Again, the laser beam was focused on the middle of the resonator, and the digital velocity signal was read from the vibrometer software in the time domain. The input signal was synchronized with a trigger signal, and the measured velocity is averaged 500 times to reduce noise.

In the vacuum tests, the vacuum level was around 7 mTorr. Unfortunately, the laser beam struggled to focus on the beam, due to additional watch glass between the microscope and the resonator. Although the relative velocity in the time domain was stable, the magnitude was dependent significantly on the focus; therefore, it was normalized to prevent any calibration error.

The data read from the LDV software is shown in Figure 9. Then, the FFT (fast Fourier transform) was applied in MATLAB^®^ to transform the data into the frequency domain.

### 5.4. Mode Shape

The mode shape of the built-in chip was measured with resonance actuation to compare the results with the models.

## 6. Results and Discussion

The results of the measurements described in Section 5 are presented and discussed in this section.

### 6.1. Frequency Sweep

The results of the frequency sweep are plotted in Figure 10. The Lorentzian-like distributions in both plots represent the RMS velocity, while the rest represent phase behavior. The horizontal axis is the actuation frequency domain, while the vertical axis for Gaussian-like behavior curves is velocity and the other is the phase axis. The resonance frequencies, $f_{r}$, are determined from the peaks of RMS velocity curves. Then, the quality factor from RMS velocity, $Q_{\mathrm{BW}}$, is found by the bandwidth, BW. Finally, the quality factor from phase slope, $Q_{\phi }$, is calculated by the phase slope at the resonance frequency, $\frac{d\phi }{\mathrm{df}}{|}_{f_{r}}$. The exact relationships are given in Equation (5).
(5)$$Q_{\mathrm{BW}}=\frac{f_{r}}{\mathrm{BW}}{\;,\;Q}_{\phi }=\frac{f_{r}}{2}\frac{d\phi }{\mathrm{df}}{|}_{f_{r}}$$

The terms shown in Equation (5) for each measurement are listed in Table 2.

The resonance frequencies, bandwidths, phase slopes, quality factor calculated from bandwidth, and quality factor calculated from phase slop are in great agreement with the reference measurement of each chip. In fact, the similarity between different devices shows the repeatability of the fabrication, while the high-quality factors point out minimal support and air damping losses with successful fabrication. These measurements clearly indicate the independency of the interconnection method on the results. Therefore, the built-in packaging offers the equivalent actuation performance as the wire bonding packaging and reference measurements with micromanipulators.

At resonance, the power supplied by the driving force (i.e., the product of the electrostatic force and velocity) is always positive, and hence, the system response (i.e., the amplitude of the velocity) is maximal.

### 6.2. Pulse Response

The pulse response results are shown in Figure 11a. Again, the horizontal axis is the actuation frequency domain, while the vertical axis represents the RMS velocity. The resonance frequencies, $f_{r}$, are determined from the peaks of RMS velocity curves. Then, the quality factor from RMS velocity, $Q_{\mathrm{BW}}$, is found by the bandwidth, BW, as illustrated in Equation (5).

The resonance frequency, bandwidth, and quality factor obtained from the bandwidth of each measurement are listed in Table 3.

The outcomes of pulse response match with the frequency sweep measurements. In addition to air tests, very high-quality factors obtained under vacuum conditions indicate very low structural losses. It should be noted that the vacuum resonance frequencies and quality factors tended to increase with time. Unfortunately, the wire bonding chip stayed under the vacuum for a shorter time, causing lower resonance frequency and quality factor. Therefore, it is not fair to compare the vacuum performances of devices. The main purpose of the vacuum test is to illustrate the application of built-in packaging inside isolated environments, and that was achieved by calculating the vacuum quality factor.

### 6.3. Mode Shape

The experimental mode shape obtained with built-in packaging is shown in Figure 12, as well as the theoretical mode shape.

The matching mode shapes demonstrate the similarity between the experiment and models. The slight variation between the theory and experiment around the edges could be sourced from the finite spot size of the laser beam causing an average measurement instead of a point measurement.

## 7. Conclusions

The main aim of this work is to propose the built-in packaging idea by illustrating its application on a single terminal device. Thus, a single-terminal actuator is designed, fabricated, packaged, and characterized with a detailed procedure.

To compare the performance of the proposed package, an actuator was designed, and a set was fabricated. Two structures were modeled with different mathematical models, FEM and ROM, for design purposes and failure analysis. Afterward, two similar actuators were packaged with the proposed method and a conventional method, wire bonding. The slight variation between devices was removed with reference measurements using micromanipulators. First, frequency sweep analysis was conducted with each device to characterize frequency response and phase behavior. Then, to consolidate the results, pulse response behavior in the frequency domain was analyzed. Outcomes of both measurements offer identical actuation performance between built-in and wire bonding packaging types. The matching results and high-quality factors obtained under air and vacuum verify the repeatability and success of the fabrication. Additionally, the compatibility of built-in packaging for operations inside an isolated chamber was demonstrated with vacuum tests. Finally, a mode shape measurement was executed using built-in packaging to justify the mathematical models. Consequently, the built-in packaging can be considered an alternative method in single-terminal device applications.

The main advantage of built-in packaging is simplicity. Primarily, no necessity for packaging equipment makes the process easy, cheap, and accessible. In addition, no chip carrier requirement gives freedom in chip design and offers immediate characterization after fabrication. Furthermore, it is a standalone packaging allowing connection to testing equipment without any PCB or cable. In addition, built-in packaging can revolutionize regular chips into plug-and-play devices. When all these benchmarks are considered, the proposed conception can catch industrial and academic attention.

Feedthrough losses and noises have a noteworthy role in the electrical actuation and measurements of MEMS and NEMS. These can be minimized by eliminating all possible interconnection elements and wires. The continuation of this work will be followed by the demonstration of single or multiple terminals to show equivalent or improved device performances.

## Figures and Tables

**Figure 1 sensors-22-05264-f001:**
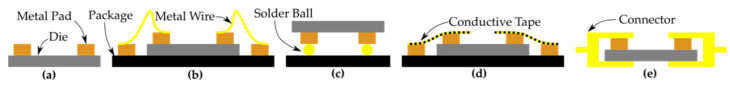
Packaging methods: (**a**) no packaging, (**b**) wire bonding, (**c**) flip-chip bonding, (**d**) tape-automated bonding, and (**e**) built-in packaging.

**Figure 2 sensors-22-05264-f002:**
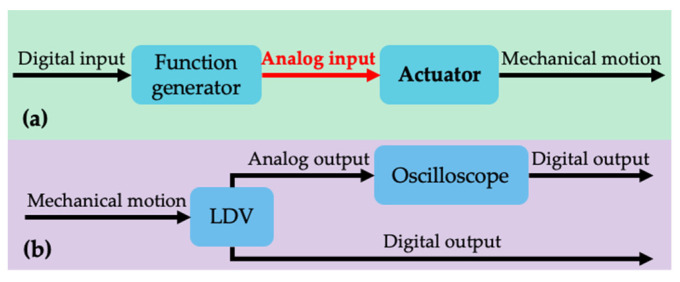
Schematics of the (**a**) actuation, and (**b**) characterization strategies. The red color indicates the focus of this study while the bold text indicates elements fabricated in the scope of this work. LDV stands for laser Doppler vibrometer.

**Figure 3 sensors-22-05264-f003:**
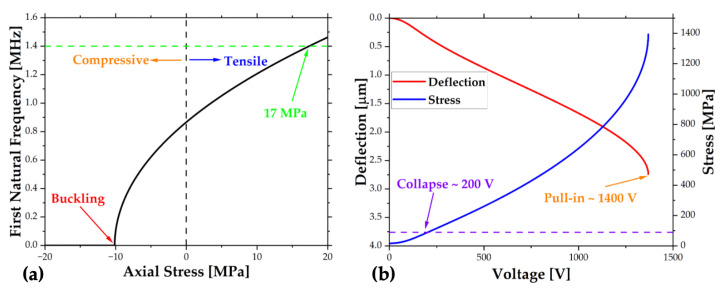
Simulation plots of the (**a**) first natural frequency vs. axial stress and (**b**) deflection and stress with respect to the actuation signal.

**Figure 4 sensors-22-05264-f004:**
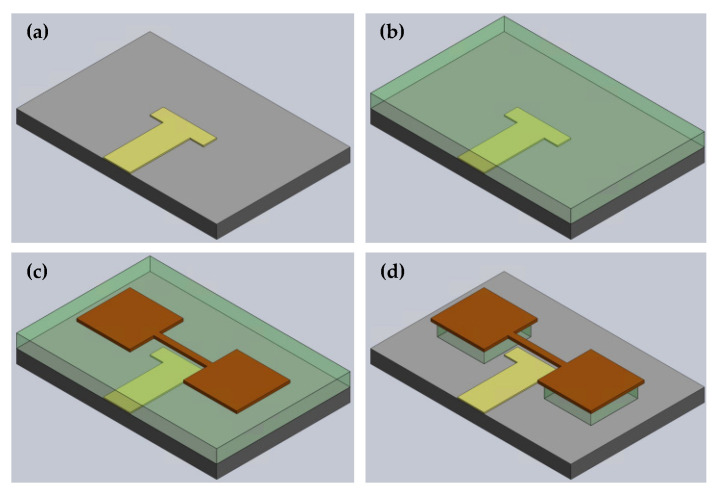
Device fabrication: (**a**) Actuation electrode patterning, (**b**) Sacrificial layer deposition, (**c**) Structure patterning, and (**d**) Releasing the resonator. (Yellow: Actuation electrode; Brown: Ground electrode and resonator; Gray: Substrate; Green: Sacrificial layer.)

**Figure 5 sensors-22-05264-f005:**
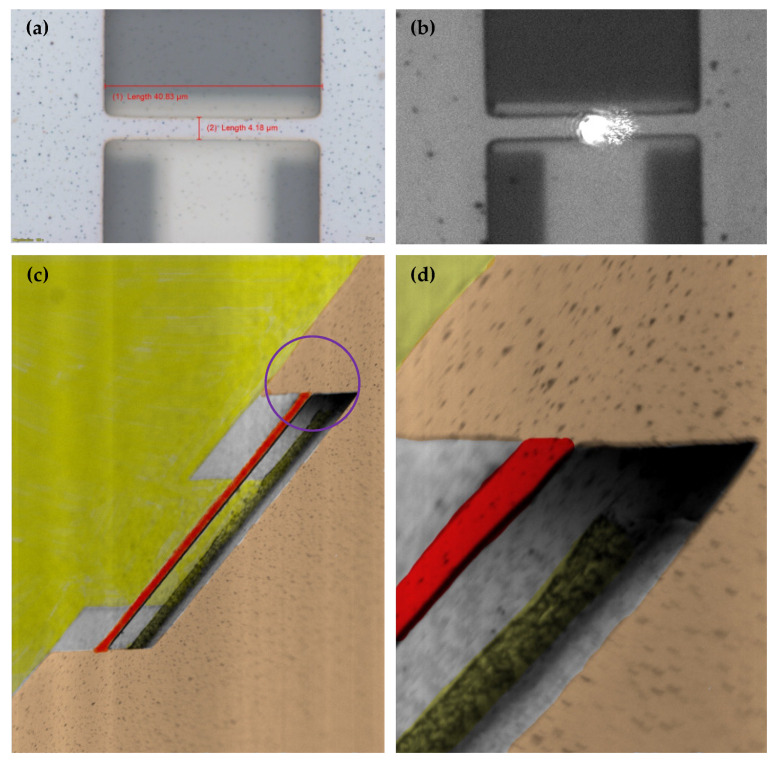
Fabricated resonator images from the following: (**a**) 100× optical microscope, (**b**) 50× optical microscope of the LDV with the laser spot, (**c**) 2000× SEM, and (**d**) 8000× SEM zoomed showing the purple area circled in (**c**). (Yellow: Actuation electrode; Brown: Ground electrode; Red: Resonator; Gray: Substrate).

**Figure 6 sensors-22-05264-f006:**
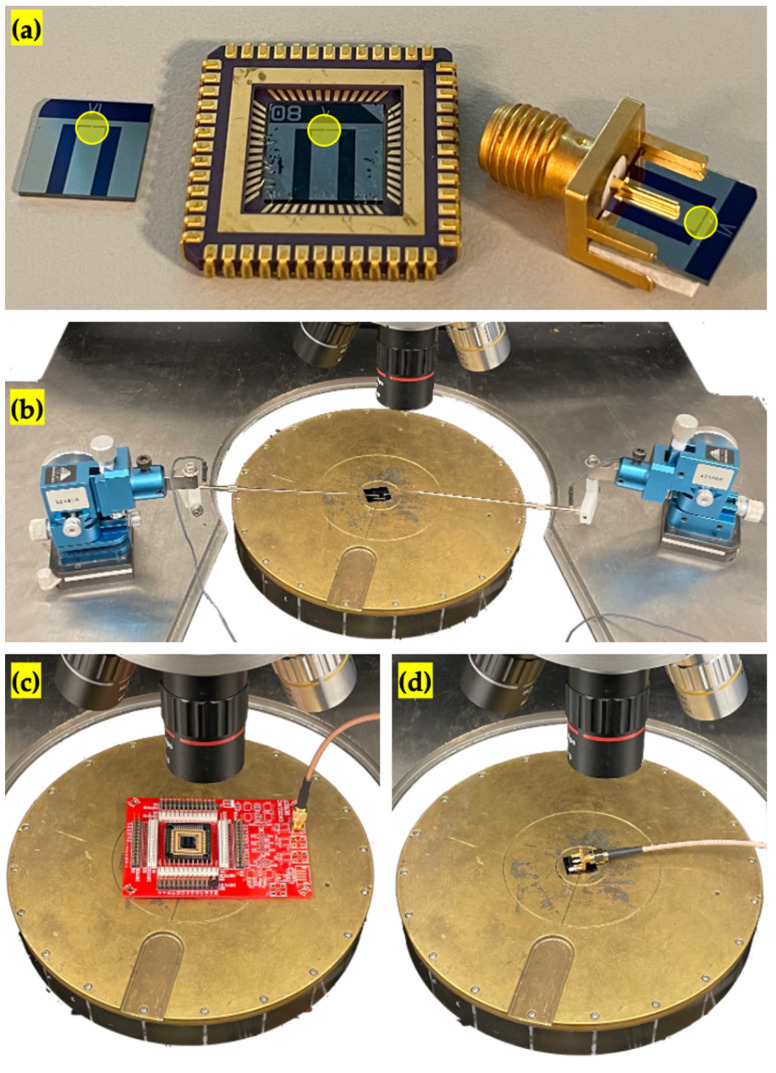
(**a**) The fabricated die without packaging, with wire bonding packaging, and with built-in packaging, from left to right. The resonator location is circled with yellow. The experimental setup comprises (**b**) reference measurements, (**c**) wire bonding packaging measurements, and (**d**) built-in packaging measurements.

**Figure 7 sensors-22-05264-f007:**
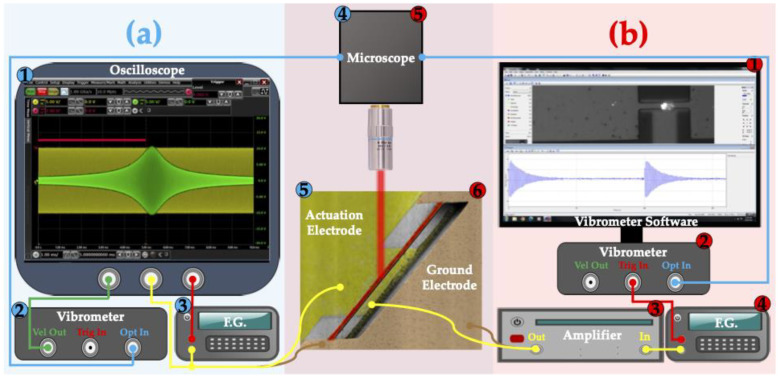
Experimental setup of (**a**) frequency sweep, and (**b**) pulse response analysis shown with blue and red backgrounds, respectively. Equipment configuration in frequency sweep analysis, (**a**), is 1: Oscilloscope, 2: Vibrometer, 3: F.G. (Function Generator), 4: Microscope, and 5: Resonator, while in pulse response analysis, (**b**), the configuration is 1: LDV software, 2: LDV, 3: Amplifier, 4: F.G., 5: Microscope, and 6: Resonator. Optical, velocity, actuation, ground, and trigger signals are shown with blue, green, yellow, brown, and red, respectively.

**Figure 8 sensors-22-05264-f008:**
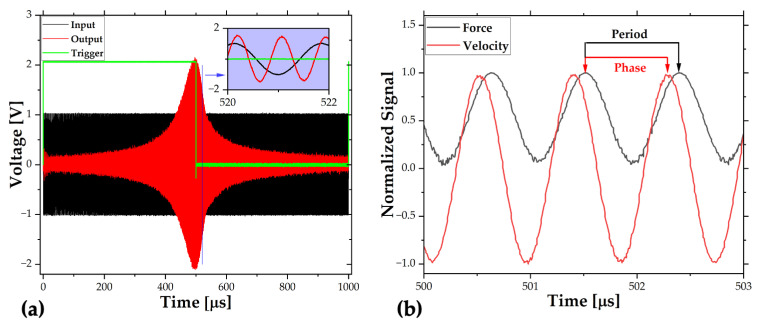
Frequency sweep measurement: (**a**) oscilloscope data, and (**b**) processes data.

**Figure 9 sensors-22-05264-f009:**
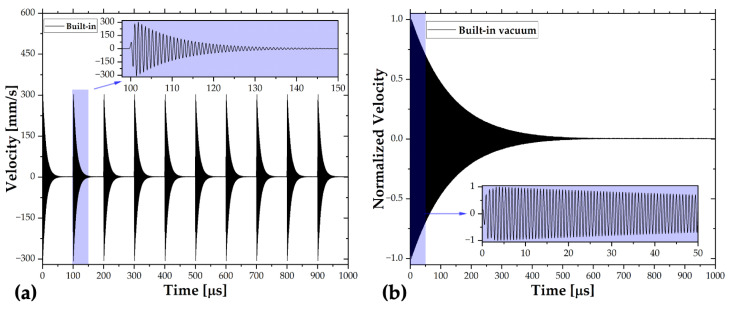
Pulse response measurement: (**a**) velocity data, and (**b**) vacuum data.

**Figure 10 sensors-22-05264-f010:**
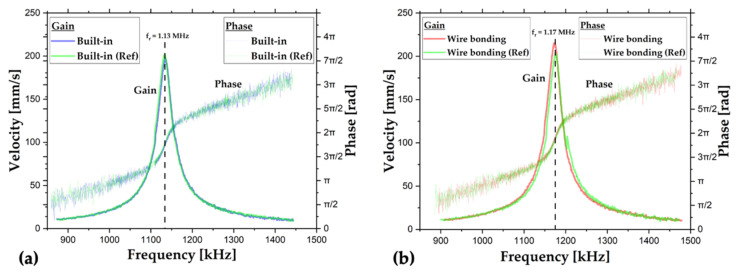
The results of frequency sweep measurements for (**a**) built-in and (**b**) wire bonding chips.

**Figure 11 sensors-22-05264-f011:**
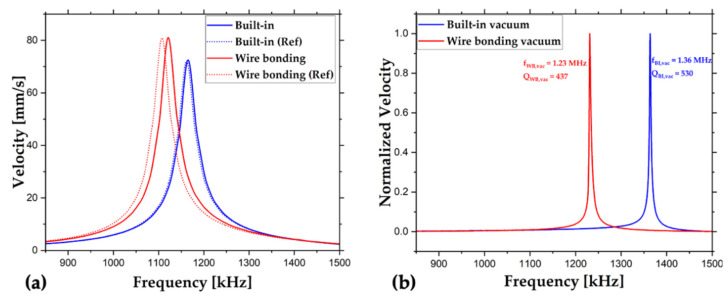
The plots of pulse response measurements performed in (**a**) in air, and (**b**) under vacuum conditions.

**Figure 12 sensors-22-05264-f012:**
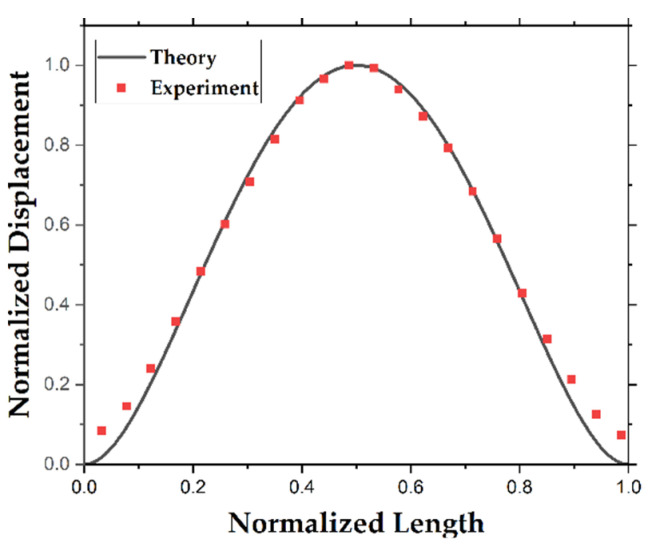
The first theoretical and experimental mode shapes.

**Table 1 sensors-22-05264-t001:** Device fabrication.

Layer	Step 1	Step 2	Step 3	Step 4
Actuation electrode	PR coating	Lithography	Metal deposition	Liftoff
Sacrificial layer	PR coating	-	-	-
Structure	Metal deposition	PR coating	Lithography	Dry etch
Dicing and Release	Dicing	Dry etch	-	-

**Table 2 sensors-22-05264-t002:** The results of frequency sweep measurements.

Device	$f_{r}\;[\mathrm{MHz}]$	BW [kHz]	$\frac{d\phi }{\mathrm{df}}\;[\mathrm{rad}/\mathrm{MHz}]$	$Q_{\mathrm{BW}}$	$Q_{\phi }$
Built-in	1.13	29.8	67.0	38.0	38.0
Built-in (Ref)	1.13	30.0	66.9	37.8	38.0
Wire bonding	1.17	30.1	67.4	38.9	39.6
Wire bonding (Ref)	1.17	28.9	67.8	40.6	39.9

**Table 3 sensors-22-05264-t003:** The results of pulse response measurements.

Device	$f_{r}\;[\mathrm{MHz}]$	BW [kHz]	$Q_{v}$
Built-in	1.17	30.1	38.7
Reference	1.16	30.2	38.4
Vacuum	1.36	2.58	529
Wire bonding	1.12	30.3	37.0
Reference	1.11	30.4	36.4
Vacuum	1.23	2.82	436

## Data Availability

Not applicable.

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
