# Peer review of "Built-In Packaging for Single Terminal Devices"

_sensors, 2022, doi:10.3390/s22145264_

Round 1

Reviewer 1 Report

Fabrication and test of a package-less actuator is described not requiring conventional interconnection elements and boards. For comparison and evaluation of the proposed method a similar devices mounted on a custom PCB and connected using wire bonding was processed simultaneously. Actuator design, function, fabrication process and measurement procedures are presented carefully in much detail. The paper is concise and should be published after some improvements detailed below:

1)     There is nothing visible in Fig. 6(c). This would be essential, since here the built-in package should appear.

2)     Labels in the figures are very small, especially in Figs. 10 and 11. They need to be increased. The representation of Fig. 2 should be improved.

3)     Examples for the adverse effect of feedthrough in MEMS resonant sensors was described in Yuanjie Xu et. al., “Single-Device and On-Chip Feedthrough Cancellation for Hybrid MEMS Resonators”, in IEEE Transactions on Industrial Electronics, vol. 59, no. 12, pp. 4930-4937, Dec. 2012, doi: 10.1109/TIE.2011.2180274, and A. Setiono et al., “Performance of an Electrothermal MEMS Cantilever Resonator with Fano-Resonance Annoyance under Cigarette Smoke Exposure”, Sensors 21 (2021) 4088 (19pp); doi: 10.3390/s21124088, which should be mentioned in the Introduction.

4)     On page 3, line 95: “The laser spot has a finite diameter called (should be skipped?) and should fully cover the top surface of the resonator …”

5)     On page 5, line 190: Figures should be numbered according to the order they appear in the text.

6)     On page 5, chapter 3: The fabrication procedures of the actuation and function layers are equal or at least very similar. Therefore, the describing text should be shortened accordingly. A table where all the details are collected would be helpful for the reader.

7)     On page 5, lines 206 and 213, Typo: Angstrom instead of Ansgtrom.

8)     On page 5, line 208: IPA should be defined.

9)     Page 7, line 248: SMA should be defined.

10)   In the caption of Fig. 7, the connection in brown should be defined (to ground electrode?).

11)   On page 10, line 327 you mean Lorentzian-like shape instead of Gaussian-like shape?

Author Response

Thanks for your detailed and constructive feedback. I'm mo

1) There is nothing visible in Fig. 6(c). This would be essential, since here the built-in package should appear.

  • Figure 6 has been updated. To demonstrate the proposed method, its implication in chip level and experimental setup are included in Figure 6, separately. The caption of the figure and some text in Section 4, where the figure is explained, are also updated.

2)  Labels in the figures are very small, especially in Figs. 10 and 11. They need to be increased. The representation of Fig. 2 should be improved.

  • Label sizes in figures 3, 8, 9, 10,11, and 12 are increased. The background color of figure 2 was missing due to Windows/macOS incomparability, and it's updated.

3)Examples for the adverse effect of feedthrough in MEMS resonant sensors was described in Yuanjie Xu et. al., “Single-Device and On-Chip Feedthrough Cancellation for Hybrid MEMS Resonators”, in IEEE Transactions on Industrial Electronics, vol. 59, no. 12, pp. 4930-4937, Dec. 2012, doi: 10.1109/TIE.2011.2180274, and A. Setiono et al., “Performance of an Electrothermal MEMS Cantilever Resonator with Fano-Resonance Annoyance under Cigarette Smoke Exposure”, Sensors 21 (2021) 4088 (19pp); doi: 10.3390/s21124088, which should be mentioned in the Introduction.

  • They're added into the introduction.

4) On page 3, line 95: “The laser spot has a finite diameter called (should be skipped?) and should fully cover the top surface of the resonator …”

  • Updated as "... diameter, called airy disk, and ...".

5) On page 5, line 190: Figures should be numbered according to the order they appear in the text.

  • It's updated.

6)   On page 5, chapter 3: The fabrication procedures of the actuation and function layers are equal or at least very similar. Therefore, the describing text should be shortened accordingly. A table where all the details are collected would be helpful for the reader.

The text is shortened and a table is added.

7) On page 5, lines 206 and 213, Typo: Angstrom instead of Ansgtrom.

  • It's removed for shortening the text.

8) On page 5, line 208: IPA should be defined.

  • It's removed for shortening the text.

9) Page 7, line 248: SMA should be defined.

  • It's defined.

10) In the caption of Fig. 7, the connection in brown should be defined (to ground electrode?).

  • It's defined.

11)  On page 10, line 327 you mean Lorentzian-like shape instead of Gaussian-like shape?

  • Yes, it's updated.

Reviewer 2 Report

The technology of the built-in packaging is briefly described and the image in Figure 6c is not visible. Thus the main technology cannot be understood by the reader.

The authors should improve the technical description of 4.2 and add a proper image to Fig. 6c

Author Response

Figure 6 and Section 4.2 are updated to improve the understanding by the reader.

Reviewer 3 Report

This paper proposes a built-in packaging. There are some problems, as described here.

1.      Manual operation, not suitable for miniaturized devices.

2.      Figure 6 (c) has no valuable information.

3.      No research or applicable value.

The reasons are as follows.

1. The packaging scheme proposed in this paper is to clip the chip into the electrode frame. There is no detailed description about the packaging method in this paper. It can only be inferred that it is completed by manual operation. So it is incompatible with the current semiconductor process. 

2. There is no picture of the sample after built-in packaging.  The figure 6(C) is blanket so it cannot give effective information. It is recommended to adjust the picture size so it can give detailed information. 

3. Too much content in the article is mostly comes from the elementary knowledge. It introduces many useless details which is not suitable for journal articles. 

4. Compared with the conventional methods, the experimental results have no advantages.

Author Response

Thanks for your important feedback. First, I just want to re-emphasize the main summary of this work.

Fabrication and test of a package-less actuator is described not requiring conventional interconnection elements and boards. For comparison and evaluation of the proposed method a similar devices mounted on a custom PCB and connected using wire bonding was processed simultaneously. Actuator design, function, fabrication process and measurement procedures are presented carefully in much detail.

1) Manual operation, not suitable for miniaturized devices.

For mechanical actuators, there're two common interconnection techniques that are probing and wire bonding. Probing is a manual operation and is suitable for miniaturized devices. However, it has its own drawbacks ie. the probes are acting as antennas and causing feedthrough between actuation and sensing signals. On the other hand, wire bonding is a well-standardized operation and is also suitable for miniaturized devices. Just like probes, microwires act as antennas and cause feedthrough. Furthermore, wire bonding requires a specialized wire bonding tool and a skilled/qualified user for the well-standardized operation. In addition to that, a chip carrier and testing board connecting the chip carrier to the testing equipment are necessary for this method.

Compared to these two methods, built-in packaging is simpler, cheaper, and more compact. I believe that the suitability of the proposed method for miniaturized devices is clearly demonstrated by characterizing a 40um x 4um x 250nm microbeam. Moreover, by eliminating the usage of standard chip carriers in industrial applications, chip/die sizes can be even miniaturized.

2) Figure 6 (c) has no valuable information.

It's updated.

3) No research or applicable value.

Thanks for your feedback. 

1) The packaging scheme proposed in this paper is to clip the chip into the electrode frame. There is no detailed description about the packaging method in this paper. It can only be inferred that it is completed by manual operation. So it is incompatible with the current semiconductor process. 

The first illustration of this concept is done in this manuscript with a mechanical resonator. Therefore, I found the incompatibility with the current semiconductor process feedback irrelevant to the overall flow of this work. Although this is out of the discussion of the work, I want to discuss why I don't agree with you about that comment. For the electrical characterization of diodes (especially we work with MIM-diodes) and transistors (especially we work with FETs), probing is the most common interconnection technique. We characterized MIM-diodes and FETs with built-in packaging in our labs by directly plugging the chip into an SMU without using a probe station. From the research point of view, removing the probe station and micromanipulator is a huge advantage because it allows characterizing semiconductors with only SMU. From the industrial point of view, removing the probe station and micromanipulator is another huge advantage because FET-based sensors can be directly operated by plugging into an SMU. Therefore, the proposed method is compatible with the current semiconductor process, especially in terms of sensing applications. 

2) There is no picture of the sample after built-in packaging.  The figure 6(C) is blanket so it cannot give effective information. It is recommended to adjust the picture size so it can give detailed information. 

The figure is updated and, picture sizes are adjusted to give detailed information.

3) Too much content in the article is mostly comes from the elementary knowledge. It introduces many useless details which is not suitable for journal articles. 

The language and flow of this work are tried to be kept easy to understand by introducing all the relevant details in a simple manner. I agree that some details are elementary knowledge for experienced readers. Section 3, Fabrication section, is shortened accordingly. However, the remaining is necessary for a clear demonstration of the proposed concept.

4) Compared with the conventional methods, the experimental results have no advantages.

It's cheaper, simpler, and more compact without sacrificing any performance parameters.

Round 2

Reviewer 2 Report

The improvements of the authors explained the packaging approach, which is the core technology for the expermental results.

Citations are clearly indicated.